# Reactogenicity of mRNA- and Non-mRNA-Based COVID-19 Vaccines among Lactating Mother and Child Dyads

**DOI:** 10.3390/vaccines10071094

**Published:** 2022-07-08

**Authors:** Beth Jacob-Chow, Kandarpa Lakshmi Vasundhara, Hon Kit Cheang, Le Ye Lee, Jia Ming Low, Zubair Amin

**Affiliations:** 1Yong Loo Lin School of Medicine, National University of Singapore, Singapore 117597, Singapore; beth.nicole@u.nus.edu (B.J.-C.); vasundhara.kandarpa@u.nus.edu (K.L.V.); le_ye_lee@nuhs.edu.sg (L.Y.L.); paeza@nus.edu.sg (Z.A.); 2Department of Paediatrics and Neonatology, Hospital Lam Wah Ee, George Town 11600, Malaysia; cheanghonkit@yahoo.com; 3Department of Neonatology, Khoo Teck Puat-National University Children’s Medical Institute, National University Health System, 1E Kent Ridge Road, NUHS Tower Block, Level 12, Singapore 119228, Singapore

**Keywords:** COVID-19 vaccines, SARS-CoV-2, lactation, mother–child dyads, reactogenicity

## Abstract

The aims of the study are to: (a) Describe the reactogenicity of WHO-approved two mRNA (Pfizer-BioNTech, Moderna) and two non-RNA (Oxford-AstraZeneca, Sinovac) vaccines among lactating mother and child pairs, and (b) Compare and contrast the reactogenicity between mRNA and non-mRNA vaccines. A cross-sectional, self-reported survey was conducted amongst 1784 lactating women who received COVID-19 vaccinations. The most common maternal adverse reaction was a local reaction at the injection site, and the largest minority of respondents, 49.6% (780/1571), reported experiencing worse symptoms when receiving the second dose compared to the first dose. Respondents reported no major adverse effects or behavioural changes in the breastfed children for the duration of the study period. Among respondents who received non-mRNA COVID-19 vaccines, a majority reported no change in lactation, but those who did more commonly reported changes in the quantity of milk supply and pain in the breast. The more commonly reported lactation changes (fluctuations in breast milk supply quantity and pain in the breast) for the non-mRNA vaccines were similar to those of respondents who received mRNA vaccines. Our study, with a large, racially diverse cohort, further augments earlier reported findings in that the COVID-19 vaccines tested in this study did not cause any serious adverse events in our population for the duration of our survey period, although long-term effects are yet to be studied.

## 1. Introduction

Although children are mostly asymptomatic or have mild SARS-CoV-2 infections, newborns are more susceptible to severe disease [1,2]. It is thus important to protect mother–child dyads from SARS-CoV-2 infections. Maternal vaccination during lactation leads to antibody transfer through breast milk [3,4], which can provide a level of protection for breastfed children. While observational studies indicate that natural immunity may offer similar levels of protection against SARS-CoV-2 infections compared to those receiving two doses of mRNA vaccines, emerging studies are showing that a combination of a previous SARS-CoV-2 infection and a COVID-19 vaccine, otherwise known as hybrid immunity, seems to confer the greatest protection against SARS-CoV-2 infections [5].

At the start of the pandemic, countries adopted varying strategies in the roll-out of their national immunisation programs among breastfeeding mothers. These varied responses were in part due to the paucity of robust data on the safety of COVID-19 vaccines. Singapore, for example, adopted a cautious approach towards vaccination, whereby breastfeeding mothers were asked to suspend breastfeeding for up to 7 days after any COVID-19 vaccination [6]. This advisory was revised as of June 2021 to allow breastfeeding after receiving mRNA COVID-19 vaccinations. In contrast, the Malaysian Ministry of Health recommended continual breastfeeding following COVID-19 vaccinations [7].

Varying national policies and conflicting consent forms which state that breastfeeding is a contraindication for COVID-19 vaccinations [8], coupled with anecdotal reports of mastitis after mRNA COVID-19 vaccinations [9], contributed to lower vaccine acceptance rates in breastfeeding mothers.

In contrast to mRNA vaccines, there is a severe paucity of cross-sectional studies with adequate participants comparing the reactogenicity of different non-mRNA COVID-19 vaccines for breastfeeding mother–child dyads. For example, a single study of 20 mother–child dyads vaccinated with CoronaVac (produced by Sinovac) has been published which showed no adverse effects in the breastfed children [10]. Hence, it is crucial to address the information gap by researching the reactogenicity of non-mRNA vaccines and comparing mRNA and non-mRNA vaccines.

The aims of the study are two-fold: (a) Describe the reactogenicity of WHO-approved two mRNA (Pfizer-BioNTech, Moderna (Spikevax, ModernaTX, Inc.) and two non-RNA (Oxford-AstraZeneca, Sinovac (CoronaVac)) vaccines among lactating mother and child pairs, and (b) Compare and contrast the reactogenicity between mRNA and non-mRNA vaccines.

## 2. Materials and Methods

### 2.1. Design

This is a descriptive, cross-sectional, self-reported survey-based research with a non-probability sampling method conducted in Singapore and Malaysia. The National Healthcare Group’s Domain Specific Review Board (DSRB) categorised the study as exempt for the Singapore site (DSRB reference number: 2021/00708). In Malaysia, similar exempt status was approved by the Joint Ethics Committee on Clinical Studies of School of Pharmaceutical Sciences, USM—Hospital Lam Wah Ee.

### 2.2. Settings

In Singapore, 99% of new mothers attempt to breastfeed their children. By 6 months of age, 42% of Singaporean children receive some form of human milk and only 1% are exclusively breastfed [11]. Similarly, 98.1% of Malaysian mothers make at least one attempt at breastfeeding their children, but unlike Singapore, the prevalence of exclusive breastfeeding up to 6 months is much higher at 47.1% [12]. 

A majority of the vaccinations administered in the early part of the pandemic in both Singapore and Malaysia were mRNA vaccines as they were the first to achieve emergency authorisation status.

### 2.3. Sample

Lactating women ≥21 years of age who received at least one dose of any of the following WHO-approved COVID-19 vaccines (Pfizer-BioNTech, Moderna, Oxford-AstraZeneca, Sinovac) were eligible for the study. Exclusion criteria were as follows: women who did not receive any WHO-approved COVID-19 vaccination, were not breastfeeding at the time of survey, could not provide consent in English, mentally disabled or mothers of premature infants born <37 weeks gestation. The survey was distributed online through social media and advertisements.

### 2.4. Measurements

Demographics, past medical history and clinical outcomes of mother–child dyads at least 7 days after the vaccine dose were determined through a structured questionnaire designed specifically for the study. The survey instrument was developed through a multi-staged process: a preliminary draft was developed from known side effects of COVID-19 vaccines reported in pre-approval trials. Severe adverse effects were defined as cardiovascular or cerebral accidents. We also reviewed available reports of adverse events from post-approval research. The preliminary draft was further reviewed by the research team consisting of four specialist paediatricians and neonatologists. Thereafter, it was pilot-tested among three lactating mothers. The administered questionnaire is available in Appendix A
Table A1.

### 2.5. Data Collection

Survey was administered from 14 August 2021 to 5 January 2022 through an online secured, specialised electronic platform (Qualtrics, Singapore^®^). Respondents’ confidentiality was maintained as only anonymised data were collected.

### 2.6. Data Analysis

Data were analysed using descriptive statistics to calculate rate and proportions with STATA^®^ version 13, Informer Technologies, Inc. Continuous data were reported as mean and standard deviations. Discontinuous data were reported as median and interquartile range. Chi-square or Fisher exact tests assessed the association between vaccine types, dose, impact on breastfeeding and covariables. A *p*-value < 0.05 was considered significant.

## 3. Results

### 3.1. Demographics of Breastfeeding Mother–Child Dyads

Table 1 shows the demographic data of mother and child dyads. In total, 2612 responses were received, of which 828 (31.7%) responses were excluded. Reasons for exclusions were as follows: 673 had incomplete or missing data, 2 had taken vaccines that were not included in our study, 63 took the vaccine in a country other than Singapore/Malaysia, and 90 were vaccinated during pregnancy but not during the lactating period. This resulted in a total of 1784 responses for analysis.

Only 2.2% (40/1784) respondents had a prior SARS-CoV-2 infection out of which 12.5% (5/40) required intensive care admission.

### 3.2. Reactogenicity of COVID-19 Vaccines

Table 2 shows the reactogenicity of COVID-19 vaccines among mother and child dyads.

#### 3.2.1. Impact on Lactating Mothers

The most common symptom reported was a local reaction at the injection site, 66.3% (1183/1784) versus 57.7% (906/1571), after the first and second dose of COVID-19 vaccinations respectively. In total, 49.6% (780/1571) of the respondents reported experiencing worse symptoms after receiving the second dose compared to the first dose.

Other commonly reported symptoms were fatigue (32.4% (578/1784) versus 42.4% (666/1571)) and headache (22.7% (405/1784) versus 31.4% (494/1571)) for the first and second dose respectively. When analysing mRNA and non-mRNA respondents separately, this trend remained consistent. The incidences of fatigue, headache, increased tactile body temperature, enlarged lymph nodes, nausea and fever over 38 °C were significantly higher after the second dose compared to the first dose (*p* < 0.05). This trend was consistent in the subgroup analysis of the breastfeeding mothers who received mRNA COVID-19 vaccines. In contrast, mothers who received non-mRNA vaccines reported fewer side effects (i.e., significant local reactions, fatigue, headache, increased tactile body temperature, lymph node reaction, nausea and fever (*p* < 0.05)) when receiving their second dose compared to the first dose (Table 2).

Some other minor and less common post-vaccination reactions were also reported by 4.4% (79/1784) and 8.5% (133/1571) respondents following the first and second dose respectively. These reactions included body aches, bloating, chills, giddiness, insomnia, joint pains, menstrual disturbances, palpitations for less than an hour, tingling limbs, and sore throat. In total, 1.1% (19/1784) and 1.0% (16/1571) of respondents reported allergy-related symptoms (i.e., hives, itchy eyes) after the first and second dose respectively. Reassuringly, no respondent developed anaphylaxis or other severe adverse reactions such as myocarditis or cerebrovascular accidents.

#### 3.2.2. Impact on Lactation

A total of 11.9% (213/1784) of respondents received one dose while 88.1% (1571/1784) of respondents received two doses of COVID-19 vaccination while breastfeeding. About half (51.7%; 923/1784) of respondents did not experience any change in lactation outcomes, and the remaining 48.3% (861/1784) of respondents reported a change in lactation outcomes. The most common reported change was related to breast milk supply. In total, 11.7% (25/213) and 13.6% (214/1571) of respondents reported a reduction in breast milk supply following one and two doses respectively, whereas an increase in breast milk supply was reported by 12.7% (27/213) and 11.8% (186/1571) of those who received one or two doses of vaccines respectively. Multivariate analysis shows that mothers with younger children were less likely to report higher breast milk production with an odds ratio of 0.92 (95% CI, 0.89 to 0.95; *p* < 0.05). Exclusively breastfeeding mothers were more likely to report a reduction in breast milk supply with an odds ratio of 1.24 (95% CI, 1.06 to1.47; *p* < 0.05) and less likely to report an increase in breast milk supply (odds ratio of 0.84; 95% CI, 0.71 to 0.98; *p* < 0.05). When adjusting for the mother’s age, child’s age, number of vaccine doses, type of vaccine and intensity of breastfeeding, the reduction in breast milk was significantly related to the younger age of the mother (*p* < 0.05) and exclusive breastfeeding habits (*p* < 0.05). Meanwhile, higher breast milk production was associated with higher maternal age (*p* < 0.05) and higher age of the child (*p* < 0.05). A minority of the respondents (1.4%; 25/1784) reported other less common side effects including the sensation of clogged ducts, thicker breast milk consistency and variations in breast milk supply (e.g., decrease in milk supply after dose one followed by an increase in supply after dose two). In total, 2.3% (5/213) and 2.4% (37/1571) reported a change in milk colour following one or two doses respectively.

Respondents who received non-mRNA vaccines were less likely to report any change in milk supply (64.8% non-mRNA versus 47.5% mRNA, *p* < 0.05) or soreness in the breasts (4.6% mRNA versus 0.9% non-mRNA, *p* < 0.05) compared to those who received mRNA vaccines (*p* < 0.05). Respondents who exclusively breastfed were more likely to report breast engorgement and changes in breast milk supply (*p* < 0.05). The length of reported symptoms related to lactation was median 3 days, IQR 3 days. A total of 1.2% (21/1784) reported that their breast milk supply was perpetually altered by the vaccine, out of which 38.1% (8/21) reported an increase, while 61.9% (13/21) experienced a reduction.

In total, 6.2% (110/1784) of respondents stopped breastfeeding for at least 24 hours after a COVID-19 vaccination dose. Far more respondents who received mRNA vaccines stopped breastfeeding, 97.2% (107/110) compared to 2.8% (3/110) of those who received non-mRNA vaccines (*p* < 0.05). The reasons stated included advice from medical experts (26.4%; 29/110), abundance of caution (25.5%; 28/110), limited availability of clinical research data (12.7%; 14/110) and fear of potential negative impact on the breastfed child (12.7%; 14/110), while 22.7% (25/110) did not specify any reason for the cessation of breastfeeding.

In addition, 8.0% (143/1784) of respondents expressed and discarded their breast milk for a mean (SD) of 7.5 (5.5) days following vaccination. A total of 0.8% (14/1784) chose to discard the first pump of expressed breast milk after their COVID-19 vaccination and then continue to breastfeed as per normal.

#### 3.2.3. Impact on Child

There were no major reported adverse effects in the breastfed children of mothers who received the COVID-19 vaccination. The more common symptoms reported were rashes (6.6% (14/213) versus 2.2% (34/1571)), diarrhoea (1.9% (4/213) versus 2.0% (32/1571)), fever (1.9% (4/213) versus 2% (33/1571)) and runny nose (2.8% (6/213) versus 1.2% (19/1571)) for mothers receiving one and two doses of vaccinations respectively. There was no significant difference in child outcomes when comparing mRNA and non-mRNA vaccine cohorts.

Mothers were also asked to describe any other possible symptoms, including changes in behaviour such as increased fussiness or tiredness, they observed in their breastfed children following their COVID-19 vaccination doses. These results were then manually coded and categorised to generate the data shown in Table 2. In total, 92.9% (1658/1784) of these breastfed children had no reported changes in behaviour after their mothers received COVID-19 vaccine. The more commonly reported behavioural symptoms were being sleepier than usual (2.8% (6/213) versus 2.4% (38/1571)) and increased fussiness (3.3% (7/213) versus 2.8% (44/1571)) following the first and second dose respectively. mRNA-vaccinated mothers were more likely to report any change in their child’s behaviour (*p* < 0.05). When accounting for the number of vaccine doses, there was no significant difference in the behavioural outcomes. Of the 126 respondents who reported changes in their child’s symptoms, 11.9% (15/126) acknowledged that their assessment of their child’s behavioural change might be confounded by other events such as teething, intercurrent illness, administration of other routine childhood vaccinations, etc.

#### 3.2.4. Subgroup Analysis of mRNA Vaccines

Table 3 shows selected maternal and child outcomes among mRNA vaccines recipients.

In total, 1539 participants (86.2%, 1539/1784) received at least one dose of mRNA vaccines, of which 91.6% (1409/1539) received Pfizer-BioNTech and 8.4% (130/1539) received Moderna vaccines. When comparing the differences in local adverse effects in mothers after receiving Pfizer-BioNTech or Moderna vaccines, there were some notable differences. For both dose one and dose two, Moderna-vaccinated mothers reported a significantly higher incidence of local reaction at the injection site (dose one: 83.1% Moderna versus 71.1% Pfizer-BioNTech, *p* < 0.05, dose two: 80.4% Moderna versus 65.2% Pfizer-BioNTech, *p* < 0.05). Following dose one alone, Moderna vaccine recipients had a higher incidence of regional lymph node enlargement (Moderna 8.5% versus Pfizer-BioNTech 3.1%., *p* < 0.05). Following dose two alone, Moderna vaccine recipients reported a higher incidence of increased body temperature (Moderna 46.4% versus Pfizer-BioNTech 26.9%, *p* < 0.05), headache (Moderna 49.1% versus Pfizer-BioNTech 34.4% *p* < 0.05), fatigue (Moderna 64.3% versus Pfizer-BioNTech 47.1%, *p* < 0.05) and fever over 38 °C (Moderna 33.9% versus Pfizer-BioNTech 9.3%, *p* < 0.05).

However, there were no significant differences in any lactation outcomes between mothers who received Pfizer-BioNTech or Moderna vaccinations. Significantly more children of Moderna-vaccinated mothers had runny noses (Moderna 25% versus Pfizer-BioNTech 18% *p* < 0.05) but there were no other significant differences. There were no significant differences in child behaviour reported between Moderna- and Pfizer-BioNTech-vaccinated mothers.

#### 3.2.5. Subgroup Analysis of Non-mRNA Vaccines

Table 3 shows selected maternal and child outcomes among non-mRNA vaccine recipients.

Among the 13.7% (245/1784) respondents who received non-mRNA COVID-19 vaccinations, the commoner symptoms were an increase in milk supply (16.7% (3/18) versus 11.9% (27/227)), decrease in milk supply (5.6% (1/18) versus 8.8% (20/227)) and pain in the breast (11.1% (2/18) versus 4.4% (10/227)) following the first and second dose respectively. Less common symptoms reported were change in milk colour (0.4% (1/227)) and local lymph node swelling (1.3% (3/227)) respectively. When comparing lactation outcomes between respondents who received the Sinovac and Oxford-AstraZeneca vaccines, there were no significant differences. The more commonly reported lactation changes (fluctuations in breast milk supply and pain in the breast) for the non-mRNA vaccines were similar to that of respondents who received mRNA vaccines. None of the mothers who took the Oxford-AstraZeneca vaccine experienced changes in breast milk colour, local lymph node swelling or breast redness. There were no differences in symptoms experienced by breastfed children when comparing the Sinovac and Oxford-AstraZeneca vaccines.

## 4. Discussion

Prior to the development of COVID-19 vaccines, there were no reports of severe effects on the child from other routine non-live vaccines while breastfeeding [13]. However, there is a paucity of data reporting both the non-mRNA and mRNA COVID-19 vaccinations of lactating women with representation from wider geographical locations and ethnicities to further boost vaccination rates among lactating women. In particular, the data reporting side effects of non-mRNA vaccines, such as the widely used Sinovac vaccine, among lactating mother and child dyads are lacking. COVID-19 vaccines are currently not authorised to be used for infants and toddlers [14], hence passive transplacental transfer of antibody coupled with breastfeeding from immunised mothers remains the only viable option for protecting this vulnerable group. Thus, this study lends a unique perspective on the short-term COVID-19 vaccine reactogenicity by surveying a large Southeast Asian population of breastfeeding mother–child dyads, whereby about 1 in 10 (9.1%; 162/1784) received the Sinovac vaccination.

In this study, approximately 80% of respondents reported some reactions for both doses at a rate that is similar to several other studies among breastfeeding mothers [15,16,17,18]. In general, respondents receiving non-mRNA vaccines reported less frequent adverse events compared to mRNA vaccines. For example, about 50% of non-mRNA vaccine recipients did not experience any symptoms after the second dose; the corresponding number for the mRNA vaccine recipients was 10.9%. Similarly, 64.8% of the mothers did not report any change in milk supply after the second dose of non-mRNA vaccine compared to 47.8% among mRNA vaccine recipients. The reported incidence of change in child behaviour was low in our study, with over 90% of mothers reporting no change following the first or second dose of either mRNA or non-mRNA vaccines.

Our study demonstrated that non-mRNA vaccines have better reactogenicity, lactation-related outcomes and child outcomes compared to mRNA vaccines. Between the two non-mRNA vaccines studied in this paper, Oxford-AstraZeneca was associated with fewer adverse events and overall better lactational outcomes. Our study augments findings from another smaller study with 20 nursing mothers who received two doses of the Sinovac vaccine in which the mothers did not report any adverse effects in their breastfed children [10]. However, the generalisability of these findings is limited by a fewer number of non-mRNA vaccine recipients.

Symptoms reported for the breastfed children – such as increased fussiness, rashes, diarrhoea and respiratory tract symptoms – are common among young children, hence the effects reported may have no causal relationship to the COVID-19 vaccination of the breastfeeding mother. Our findings are in keeping with other reported literature whereby the majority surveyed reported no or minimal side effects among breastfed children following maternal vaccination [15,16,17].

The majority respondents reported no change to their lactation quantity or quality following vaccination. The most common effect experienced was a transient change in the quantity of breast milk supply, with increased or decreased supply reported at similar frequencies. This is in keeping with reported literature [15,16,17].

A few plausible mechanisms whereby maternal symptoms may impact lactation exist. A myriad of factors can affect breast milk supply, including general maternal health, amount of sleep and anxiety. Maternal fever could have a detrimental impact on breastfeeding by reducing the total body water available for milk production as fever increases metabolic demands [17]. Secondly, fatigue may contribute to a reduction in breast milk supply [19]. Hence, it may be helpful for breastfeeding women to be counselled on strategies to maintain their breast milk supply, particularly if they are feeling unwell following vaccination.

Limitations in our study include subjective reports from study participants; for example, there was no measurement of milk volume to quantify any perceived change in milk supply. Participants’ reported symptoms and signs were not verified by healthcare professionals. As with self-reported surveys, there is a possibility of recall bias resulting in over- or underestimation of the events reported. Our respondents’ awareness of media coverage and public opinions of adverse events following COVID-19 vaccinations—which were prominent and widely discussed during the duration of this study—may have resulted in increased rates of reports of adverse events among our population. Lastly, a limitation in our study design is the absence of a control group to provide data from unvaccinated breastfeeding mothers for inclusion and comparison.

Nevertheless, our study provides a better understanding of the effects and safety of four commonly used mRNA and non-mRNA COVID-19 vaccines among lactating mothers and their children. Our study lends further support to other published studies with more study participants and wider geographical and racial mix. In the face of potential threats from new SARS-CoV-2 variants, further studies appraising adverse events incidence in relation to booster doses and vaccine interchangeability are still needed to provide a broader safety aspect for the vaccines currently being used for this population.

## Figures and Tables

**Table 1 vaccines-10-01094-t001:** Demographic characteristics of breastfeeding mother–child dyads (*n* = 1784).

**Characteristics of respondents [*n* (%)]**	
**Mean age of mothers in years (SD)**	**32.7 +/− 3.9**
**Age range of the youngest child who was breastfed**	
<1 month	81 (4.5)
1–6 months	714 (40.0)
7–12 months	509 (28.5)
13–18 months	290 (16.3)
19–24 months	124 (7.0)
25–36 months	56 (3.1)
>36 months	10 (0.6)
**Highest education level of mother**	
Primary school	4 (0.2)
Secondary school	53 (3.0)
High school diploma or equivalent	264 (14.8)
Undergraduate degree	813 (45.6)
Postgraduate degree	604 (33.9)
Others	46 (2.5)
**Brand of vaccine**	
Pfizer-BioNTech (mRNA vaccine)	1409 (79.0)
Moderna (mRNA vaccine)	130 (7.3)
Sinovac (non-mRNA vaccine)	162 (9.1)
Oxford-AstraZeneca (non-mRNA vaccine)	83 (4.6)
**Participating country**	
Singapore	1225 (68.7)
Malaysia	559 (31.3)
**Ethnic group**	
Chinese	1078 (60.4)
Malay	496 (27.8)
Indian	62 (3.5)
Others	148 (8.3)
**COVID-19 vaccines received [*n* (%)]**	
1 dose	213 (11.9)
2 doses	1571 (88.1)
**Milk intake of the child**	
Exclusively breastfed	1010 (56.6)
Half human milk, half other liquids	336 (18.8)
Some human milk, mostly other liquids	36 (2.0)
Some human milk, mostly other solid feeds	396 (22.2)
Indeterminate	6 (0.4)
**Medical history of respondents [*n* (%)]**	
**Significant maternal past medical history**	
None	1622 (91.0)
Asthma	34 (1.9)
Cardiovascular diseases	12 (0.7)
Type 2 diabetes mellitus	8 (0.4)
Autoimmune diseases	8 (0.4)
Others	100 (5.6)
**Maternal medications**	
None	1691 (94.8)
Medications (e.g., Domperidone) to improve milk supply	4 (0.2)
Immunosuppressant	7 (0.4)
Medications for asthma	6 (0.3)
Oral contraceptive	18 (1.0)
Others	58 (3.3)
**Maternal allergy history**	
None	1427 (80.0)
Drug allergy	179 (10.0)
Food allergy	78 (4.4)
Environmental allergy	61 (3.4)
Multiple allergies	39 (2.2)

**Table 2 vaccines-10-01094-t002:** Maternal–child outcomes of respondents who received the COVID-19 vaccine(s).

Adverse Reactions among Lactating Mothers Who Received the COVID-19 Vaccine(s)									
Type of Adverse Reactions	Dose 1 mRNA [*n* (%)] *n* = 1539	Dose 2 mRNA [*n* (%)] *n* = 1344	*p* Value	Dose 1 Non-mRNA [*n* (%)] *n* = 245	Dose 2 Non-mRNA [*n* (%)] *n* = 227	*p* Value	Dose 1 Overall [*n* (%)] *n* = 1784	Dose 2 Overall [*n* (%)] *n* = 1571	*p* Value
No symptoms	256 (16.6)	147 (10.9)	<0.05	91 (37.1)	115 (50.6)	<0.05	347 (19.5)	262 (16.7)	<0.05
Reaction at the injection site	1110 (72.1)	858 (63.8)	<0.05	73 (29.8)	48 (21.1)	<0.05	1183 (66.3)	906 (57.7)	<0.05
Fatigue or low mood	503 (32.7)	627 (46.7)	<0.05	75 (30.6)	39 (17.2)	<0.05	578 (32.4)	666 (42.4)	<0.05
Headache	326 (21.2)	460 (34.2)	<0.05	79 (32.2)	34 (15.0)	<0.05	405 (22.7)	494 (31.4)	<0.05
Body temperature up to 38 °C	171 (11.1)	369 (27.5)	<0.05	47 (19.2)	8 (3.5)	<0.05	218 (12.2)	377 (24.0)	<0.05
Soreness and enlarged lymph node	55 (3.6)	95 (7.1)	<0.05	1 (0.4)	1 (0.4)	1.00	56 (3.1)	96 (6.1)	<0.05
Nausea	47 (3.1)	73 (5.4)	<0.05	14 (5.7)	3 (1.3)	<0.05	61 (3.4)	76 (4.8)	<0.05
Fever (over 38 °C)	33 (2.1)	148 (11.0)	<0.05	15 (6.1)	2 (0.8)	<0.05	48 (2.7)	150 (9.5)	<0.05
Runny nose	25 (1.6)	46 (3.4)	<0.05	6 (2.5)	1 (0.4)	0.07	31 (1.7)	47 (3.0)	<0.05
Diarrhoea	22 (1.4)	27 (2.1)	0.23	4 (1.6)	3 (1.3)	0.54	26 (1.5)	30 (1.9)	0.31
Chest pain	23 (1.5)	28 (2.1)	0.23	6 (2.5)	6 (2.6)	0.56	29 (1.6)	34 (2.2)	0.25
Cough	10 (0.7)	13 (0.9)	0.34	3 (1.2)	0 (0)	0.14	13 (0.8)	13 (0.8)	0.74
Vomiting	7 (0.5)	12 (0.8)	0.15	2 (0.8)	1 (0.4)	0.52	9 (0.5)	13 (0.8)	0.25
Allergic symptoms	17 (1.1)	12 (0.9)	0.57	2 (0.8)	4 (1.8)	0.31	19 (1.1)	16 (1.0)	0.89
Others	74 (4.8)	128 (9.5)	<0.05	5 (2.0)	5 (2.2)	0.58	79 (4.4)	133 (8.5)	<0.05
Comparison of dose one versus dose two of vaccines received									
Comparison			mRNA [*n* (%)] *n* = 1539	Non-mRNA [*n* (%)] *n* = 245	*p* value		Total [*n* (%)] *n* = 1784		
No symptoms experienced either time			74 (4.8)	73 (29.8)	<0.05		147 (8.2)		
Reaction same for both doses			248 (16.1)	41 (16.7)	0.06		289 (16.2)		
Reaction worse for first dose			231 (15.0)	64 (26.1)	<0.05		295 (16.5)		
Reaction worse for second dose			744 (48.3)	36 (14.7)	<0.05		780 (43.7)		
Not applicable (only one dose received, or unable to make a comparison)			242 (15.7)	31 (12.7)	0.22		273 (15.3)		
Lactation-related outcomes of mothers who received COVID-19 vaccine(s) [*n* (%)]									
Type of Outcome	Received only one mRNA dose [*n* (%)] *n* = 195	Received two mRNA doses [*n* (%)] *n* = 1344	*p* value	Received only one non-mRNA dose [*n* (%)] *n* = 18	Received two non-mRNA doses [*n* (%)] *n* = 227	*p* value	Received only one vaccine dose (overall) [*n* (%)] *n* = 213	Received two vaccine doses (overall) [*n* (%)] *n* = 1571	*p* value
No change in milk supply	104 (53.3)	638 (47.5)	0.13	11 (61.1)	147 (64.8)	0.76	115 (54.0)	785 (50.0)	0.27
Reduction in milk supply	24 (12.3)	194 (14.4)	0.43	1 (5.6)	20 (8.8)	0.63	25 (11.7)	214 (13.6)	0.49
Increase in milk supply	24 (12.3)	159 (11.8)	0.85	3 (16.7)	27 (11.9)	0.55	27 (12.7)	186 (11.8)	0.72
Breast engorgement	9 (4.6)	81 (6.0)	0.43	0 (0)	10 (4.4)	0.46	9 (4.2)	91 (5.8)	0.25
Change in milk colour	5 (2.6)	36 (2.7)	0.93	0 (0)	1 (0.4)	0.92	5 (2.3)	37 (2.4)	0.99
Soreness of breast	11 (5.6)	62 (4.6)	0.53	1 (5.6)	2 (0.9)	0.20	12 (5.6)	64 (4.1)	0.29
Nipple pain	9 (4.6)	72 (5.4)	0.66	2 (11.1)	10 (4.4)	0.22	11 (5.2)	82 (5.2)	0.97
Lymph node swelling at the neck or axillary areas	2 (1.0)	31 (2.3)	0.25	0 (0)	3 (1.3)	0.79	2 (0.9)	34 (2.2)	0.23
Breast redness	2 (1.0)	30 (2.2)	0.27	0 (0)	5 (2.2)	0.68	2 (0.9)	35 (2.2)	0.22
Others	5 (2.6)	41 (3.1)	0.71	0 (0)	2 (0.9)	0.86	5 (2.3)	43 (2.7)	0.74
Outcomes of breastfed children whose mothers received at least one dose of the COVID-19 vaccine while breastfeeding [*n* (%)]									
Type of child outcome	Received only one mRNA dose [*n* (%)] *n* = 195	Received two mRNA doses [*n* (%)] *n* = 1344	*p* value	Received only one non-mRNA dose [*n* (%)] *n* = 18	Received two non-mRNA doses [*n* (%)] *n* = 227	*p* value	Received only one vaccine dose (overall) [*n* (%)] *n* = 213	Received two vaccine doses (overall) [*n* (%)] *n* = 1571	*p* value
Rash	13 (6.6)	29 (2.2)	<0.05	0 (0)	5 (2.2)	0.52	14 (6.6)	34 (2.2)	<0.05
Diarrhoea	4 (2.1)	28 (2.1)	0.98	0 (0)	4 (1.8)	0.57	4 (1.9)	32 (2.0)	0.88
Fever	4 (2.1)	28 (2.1)	0.98	0 (0)	5 (2.2)	0.52	4 (1.9)	33 (2.1)	0.83
Runny nose	6 (3.1)	17 (1.3)	0.05	0 (0)	2 (0.9)	0.69	6 (2.8)	19 (1.2)	0.06
Cough	3 (1.5)	15 (1.1)	0.61	0 (0)	2 (0.9)	0.69	3 (1.4)	17 (1.1)	0.67
Vomiting	2 (1.0)	6 (0.4)	0.29	0 (0)	0 (0)	N/A	2 (0.9)	6 (0.4)	0.25
Refusal to feed	0 (0)	1 (0.1)	0.70	0 (0)	0 (0)	N/A	0 (0)	1 (0.1)	0.71
Behavioural outcomes of breastfed children whose mothers received at least one dose of the COVID-19 vaccine while breastfeeding [*n* (%)]									
Type of outcome	Received only one mRNA dose [*n* (%)] *n* = 195	Received two mRNA doses [*n* (%)] *n* = 1344	*p* value	Received only one non-mRNA dose [*n* (%)] *n* = 18	Received two non-mRNA doses [*n* (%)] *n* = 227	*p* value	Received only one vaccine dose (overall) [*n* (%)] *n* = 213	Received two vaccine doses (overall) [*n* (%)] *n* = 1571	*p* value
No significant behavioural changes	177 (90.8)	1244 (92.6)	0.38	17 (94.4)	220 (96.9)	0.57	194 (91.1)	1464 (93.2)	0.26
Sleepier than usual	6 (3.1)	36 (2.7)	0.75	0 (0)	2 (0.8)	0.86	6 (2.8)	38 (2.4)	0.73
Increased fussiness	6 (3.1)	42 (3.1)	0.97	1 (5.6)	2 (0.8)	0.21	7 (3.3)	44 (2.8)	0.69
Refusal to feed	1 (0.5)	5 (0.4)	0.56	0 (0)	1 (0.4)	0.93	1 (0.4)	6 (0.4)	0.85
Others	5 (2.6)	17 (1.3)	0.15	0 (0)	2 (0.8)	0.86	5 (2.3)	19 (1.2)	0.18

**Table 3 vaccines-10-01094-t003:** Maternal–child outcomes of respondents who received non-mRNA vaccine.

Lactation-Related Outcomes of Mothers Who Received the COVID-19 Vaccine [*n* (%)]			
Type of Outcome	Received One or Two Doses of the Sinovac Vaccine [*n* (%)] *n* = 162	Received One or Two Doses of the Oxford-AstraZeneca Vaccine [*n* (%)] *n* = 83	Comparison (*p* Value)
No change in milk supply	98 (60.4)	60 (72.3)	0.07
Reduction in milk supply	16 (9.9)	5 (6.0)	0.31
Increase in milk supply	19 (11.7)	11 (13.3)	0.73
Breast engorgement	7 (4.3)	3 (3.6)	0.54
Change in milk colour	1 (0.6)	0 (0)	0.66
Soreness of breast	2 (1.2)	1 (1.2)	0.73
Nipple pain	9 (5.6)	3 (3.6)	0.37
Lymph node swelling at the neck or axillary areas	3 (1.9)	0 (0)	0.29
Breast redness	5 (3.1)	0 (0)	0.13
Other lactation complaints	2 (1.2)	0 (0)	0.43
Outcomes of breastfed children whose mothers received at least one dose of the COVID-19 vaccine while breastfeeding [*n* (%)]			
Type of outcome	Received one or two doses of the Sinovac vaccine [*n* (%)] *n* = 162	Received one or two doses of the Oxford-AstraZeneca vaccine [*n* (%)] *n* = 83	Comparison (*p* value)
Rash	3 (1.9)	1 (1.2)	0.58
Diarrhoea	4 (2.5)	1 (1.2)	0.45
Fever	2 (1.2)	3 (3.6)	0.22
Runny nose	2 (1.2)	1 (1.2)	0.73
Cough	2 (1.2)	0 (0)	0.44
Vomiting	0 (0)	1 (1.2)	0.34
Refusal to feed	0 (0)	1 (1.2)	0.34
Behavioural outcomes of breastfed children whose mothers received at least one dose of the COVID-19 vaccine while breastfeeding [*n* (%)]			
Type of outcome	Received one or two doses of the Sinovac vaccine [*n* (%)] *n* = 162	Received one or two doses of the Oxford-AstraZeneca vaccine [*n* (%)] *n* = 83	Comparison (*p* value)
No significant behavioural changes reported	146 (90.1)	81 (97.6)	<0.05
Sleepier than usual	1 (0.6)	1 (1.2)	0.56
Increased fussiness	2 (1.2)	1 (1.2)	0.73
Refusal to feed	1 (0.6)	0 (0)	0.66
Others	2 (1.2)	0 (0)	0.44

## Data Availability

The data presented in this study are available upon request from the corresponding author.

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
