# Peer review of "Reactogenicity of mRNA- and Non-mRNA-Based COVID-19 Vaccines among Lactating Mother and Child Dyads"

_vaccines, 2022, doi:10.3390/vaccines10071094_

Round 1
Reviewer 1 Report
Reactogenicity of mRNA and non-mRNA based COVID-19 vac-2 cines among lactating mother and baby dyads
In this study, the authors surveyed lactating mothers to determine effects of COVID vaccines on milk production and quality of breastfeeding experience. Based on the data provided, it is hard for me to agree with the conclusion that COVID vaccines are safe for lactating/breastfeeding mothers, given that nearly 48% women show altered lactation or adverse events after dose 2. It is also unclear as to how long after the vaccines did these effects on altered lactation last. Since there was no control or reference group, this would not be possible to ascertain. It would be prudent to stick to data, rather than making sweeping, incorrect, and misleading statements. Overall, the conclusions are misleading at best and overstate the safety of COVID vaccines in lactating and breastfeeding women. Long-term effects of COVID vaccines in adults, children, and passing of antibodies via breastmilk to infants remains to be determined. I have several major concerns and some minor concerns with data interpretation and presentation.
Major concerns:
1. I am struck by authors’ opening sentence in the Discussion section: “There is no biological plausibility that COVID-19 vaccines would cause harm to the breastfed children as the vaccines do not contain live componentsand there is no known risk associated with being given a non-live vaccine while breastfeeding”. This is just not true. Leaving aside all the mechanistic studies that have shown that the mRNA vaccines can alter cell function at many levels, including being integrated in the genome, the statement is reflective of complete lack of vaccine action. There are no vaccines or drugs that are free of adverse effects or complications and thus it is an unscientific statement.
2. In absence of a control or a reference group (unvaccinated lactating mothers with and without COVID), I am unclear as to how the authors can conclude that the vaccines are safe? In absence of the control group, it is unclear how if greater than 30% of the mothers report some sort of adverse event, how can it be concluded that vaccines are safe? How many mothers without vaccine report severe COVID symptoms or adverse events from any other infectious disease?
3. mRNA COVID vaccine is known to increase blood viscosity and thereby alter milk supply. Is that not a concern? Please see PMID: 34142570 and PMID: 26859784
4. Reference # 17: Low et al show in their paper that in 10% of the lactating mothers, up to 2ng/mL of mRNA vaccine can be detected. If a baby drinks about 8oz of milk per feed, then per day, a child is exposed to about 1.6-2µg of mRNA per day. There is not clear understanding of how these mRNA might modulate cell function.
5. It is curious that the authors address 43.7% of the respondents as minority! This is not a small number and dismissing the fact that 43.7% reported worsening of symptoms after dose 2 as vaccines being safe is surprising. I do not agree with this conclusion. What were the symptoms that worsened? There is no link to Appendix A.
6. Statisitical tests between the vaccine groups need to be performed as well, not just between the first and second dose of vaccines (between mRNA and non mRNA vaccine groups)
7. Since mRNA vaccines caused significantly more adverse events than non-mRNA vaccines, please perform subgroup analysis on those instead of non-mRNA vaccines. Perhaps the authors performed such an analysis but failed to report as they state in section 3.2.1, “This trend was consistent in the subgroup analysis of the breastfeeding mothers who received mRNA COVID-19 vaccines”. I can only find subgroup analysis for non-mRNA vaccines, not for mRNA vaccines. Please provide complete statistical analysis.
8. It is notable that 9.5% of participants who received two mRNA doses reported “Other” adverse reactions. I am unable to find Appendix A; could some serious AEs be mischaracterized under “Others” category.
9. Please check the numbers in Table 3 (Behavioral Outcomes). n = 162 but the values add up to 152 (93.7%). What was the age of the children on which these outcomes were measured?
Minor
1. In the introduction, lines 51-52, the authors cite a publication about mastitis (reference 8) and yet call it anecdotal reports. Why are documented cases of mastitis anecdotal?
2. Lines 57-58: “For example, a single study of 20 mother child dyads vaccinated with CoronaVac (produced by Sinovac) has been published which showed no adverse effects in the breastfed children [9]”. The study has a very small sample size and the adverse events analyzed are not listed in the publication, hence how is this statement accurate?
3. How did the authors assess adverse events (AEs) in their cohort? Did they provide a list of AEs and were the participants free to report AEs not listed on the survey?
4. Others have shown that COVID vaccines induce higher antibody titers that cross the placental barrier and are found in umbilical cord blood (https://www.medrxiv.org/content/10.1101/2021.03.07.21253094v1). The effect of these antibodies cannot be ascertained in few days or week.
5. In fact, a similar study conducted in the US PMID: 34492204 seems to have similar observations as the authors present here, hence it would be nice if the authors can highlight the differences and uniqueness of their study.
6. Lines 24-27: the sentence needs to be re-written. How can the majority experience both increase and decrease in lactation?
7. It would be helpful to know exclusion and inclusion criteria.
8. It is unclear as to what questions were asked of the mothers? Was the data collected within 6 months of birth? Appendix A is nowhere to be found.
Author Response
Dear Reviewer,
Journal: Vaccines (ISSN 2076-393X)
Subject: Reactogenicity of mRNA and non-mRNA based COVID-19 vaccines among lactating mother and baby dyads” (Manuscript ID: vaccines-1746676)
We thank the reviewers for their comments and have amended the manuscript to address all their concerns accordingly. For ease of reference to the changes made, we have tabulated them below and highlighted them in our manuscript. We hope that the revised version will be suitable for publication. We look forward to hearing from you again.
Reviewer’s comment |
Response |
Section |
I am struck by the authors' opening sentence in the Discussion section: “There is no biological plausibility that COVID-19 vaccines would cause harm to the breastfed children as the vaccines do not contain live components and there is no known risk associated with being given a non-live vaccine while breastfeeding”. This is just not true. Leaving aside all the mechanistic studies that have shown that the mRNA vaccines can alter cell function at many levels, including being integrated in the genome, the statement is reflective of complete lack of vaccine action. There are no vaccines or drugs that are free of adverse effects or complications and thus it is an unscientific statement. |
Dear Reviewer, thank you for the comment. We have removed the statement and amended it such that it provides a more balanced view based on existing literature.
We have removed this line “There is no biological plausibility that COVID-19 vaccines would cause harm to the breastfed children as the vaccines do not contain live components and there is no known risk associated with being given a non-live vaccine while breastfeeding [12].” |
Discussion |
In absence of a control or a reference group (unvaccinated lactating mothers with and without COVID), I am unclear as to how the authors can conclude that the vaccines are safe? In absence of the control group, it is unclear how if greater than 30% of the mothers report some sort of adverse event, how can it be concluded that vaccines are safe? How many mothers without vaccines report severe COVID symptoms or adverse events from any other infectious disease? |
Dear Reviewer, thank you for the comment. 99% and 98.9% of adverse events reported by our participants who received 1 or 2 doses of vaccines respectively include minor, treatable symptoms such as local reaction at injection site, elevated body temperature, fatigue, headache, nausea, fever, runny nose, diarrhea, cough, vomiting, chest pain.
The only severe symptoms reported were those of allergic reaction such as hives, the incidence of which was very rare. There were 0 reports of anaphylaxis, cerebrovascular events and myocarditis. It is with this information that we concluded that the vaccines are safe for the mothers.
Due to the nature of our study design, no control group was added. This is because based on various international guidelines, including WHO, it is recommended that lactating mothers receive COVID-19 vaccinations for immunity. Hence making it ethically unfeasible for a control group to be used.
That aside, during the study conception phase, we foresaw other various other issues with including a control cohort such as finding an equal number of unvaccinated, lactating, participants, accounting various confounding variables such as age and race. |
N/A |
mRNA COVID vaccine is known to increase blood viscosity and thereby alter milk supply. Is that not a concern? Please see PMID: 34142570 and PMID: 26859784 |
Dear Reviewer, thank you for the comment and references stated. While the paper PMID: 34142570 raises a valid concern regarding blood viscosity, it is not directly related to breast milk. Moreover, reports of thicker milk consistency were few in our cohort (reported within 1.4% of “other” lactation symptoms) hence it is not clinically or statistically significant and hence was not pursued. |
N/A |
Reference # 17: Low et al show in their paper that in 10% of the lactating mothers, up to 2ng/mL of mRNA vaccine can be detected. If a baby drinks about 8oz of milk per feed, then per day, a child is exposed to about 1.6-2µg of mRNA per day. There is not clear understanding of how these mRNA might modulate cell function. |
Dear Reviewer, thank you for the comment and highlighting this as a concern.
We would like to invite you to read our paper (Reference # 3; Low, J., Gu, Y., Ng, M., Amin, Z., Lee, L., Ng, Y., Shunmuganathan, B., Niu, Y., Gupta, R., Tambyah, P., MacAry, P., Wang, L. and Zhong, Y. Codominant IgG and IgA expression with minimal vaccine mRNA in milk of BNT162b2 vaccinees. npj Vaccines 2021, 6(1).) which suggested that in most cases, vaccine mRNA does not escape into mammary secretions. The few instances where extremely low levels of BNT162b2 mRNA were detected may be due to naturally occurring inter-individual variations in protein adsorption. This miniscule amount of mRNA is expected to be readily destroyed by enzymes in the infant’s gut, and any accompanying lipid nanoparticles that are excreted into human milk would also be readily digested if ingested orally by the infant.
|
N/A |
It is curious that the authors address 43.7% of the respondents as minority! This is not a small number and dismissing the fact that 43.7% reported worsening of symptoms after dose 2 as vaccines being safe is surprising. I do not agree with this conclusion. What were the symptoms that worsened? There is no link to Appendix A. |
Dear Reviewer, thank you for the comment.
The majority of adverse events reported are minor, self-limiting symptoms. Weighing the benefits versus minor adverse effects of receiving the COVID-19 vaccine, it is safe to recommend it. These reported percentages in our study group are similar to the general population who had received COVID-19 vaccinations.
We have rephrased the sentence for clarity. “The most common maternal adverse reaction was a local reaction at the injection site; the largest minority of respondents, 43.7% (780/1784), reported experiencing worse symptoms when receiving the second dose compared to the first dose. “ This has been amended to: “The most common maternal adverse reaction was a local reaction at the injection site; the largest minority of respondents, 43.7% (780/1784), reported experiencing worse symptoms when receiving the second dose compared to the first dose.” |
Abstract |
Statistical tests between the vaccine groups need to be performed as well, not just between the first and second dose of vaccines (between mRNA and non mRNA vaccine groups) |
Dear Reviewer, thank you for the comment. We have incorporated comparison between mRNA and non-mRNA vaccines in section 3.1 and all sub sections. We have also included sub-group analysis of mRNA (section 3.2.4) and non-mRNA (section 3.2.5). |
Section 3.1, 3.2.4, 3.2.5 |
Since mRNA vaccines caused significantly more adverse events than non-mRNA vaccines, please perform subgroup analysis on those instead of non-mRNA vaccines. Perhaps the authors performed such an analysis but failed to report as they state in section 3.2.1, “This trend was consistent in the subgroup analysis of the breastfeeding mothers who received mRNA COVID-19 vaccines”. I can only find subgroup analysis for non-mRNA vaccines, not for mRNA vaccines. Please provide complete statistical analysis. |
Dear Reviewer, thank you for the comment. We identified a paucity of cross-sectional studies with adequate participants comparing the reactogenicity of different non-mRNA COVID-19 vaccines for breastfeeding mother-child dyads. This is why we specifically included the non-mRNA section for subgroup analysis.
However, we note your input that there were more adverse events among our mRNA cohort so we included the mRNA subgroup analysis in section 3.2.4. |
Section 3.2.4 |
It is notable that 9.5% of participants who received two mRNA doses reported “Other” adverse reactions. I am unable to find Appendix A; could some serious AEs be mischaracterized under “Others” category. |
Dear Reviewer, thank you for the comment. We sincerely apologize for the oversight and we have included Appendix A for your reference.
When collecting data for reactogenicity, we allowed the participants to input free text for “other” symptoms. Subsequently, 2 independent reviewers sifted through the responses to ensure consistency. |
Appendix A |
Please check the numbers in Table 3 (Behavioral Outcomes). n = 162 but the values add up to 152 (93.7%). What was the age of the children on which these outcomes were measured? |
Dear Reviewer, thank you for the comment. We have reviewed these numbers. The age range of children in our study is 0-39 months. |
Table 2 |
In the introduction, lines 51-52, the authors cite a publication about mastitis (reference 8) and yet call it anecdotal reports. Why are documented cases of mastitis anecdotal? |
Dear Reviewer, thank you for the comment.
We have clarified by stating it more explicitly by removing the word ‘anecdotal’.
Amended from: “Varying national policies and conflicting consent forms which state that breastfeeding is a contraindication for COVID-19 vaccinations [7], coupled with anecdotal reports of mastitis after mRNA COVID-19 vaccinations [8] contributed to lower vaccine acceptance rates in breastfeeding mothers.” To
“Varying national policies and conflicting consent forms which state that breastfeeding is a contraindication for COVID-19 vaccinations [7], coupled with reports of mastitis after mRNA COVID-19 vaccinations [8] contributed to lower vaccine acceptance rates in breastfeeding mothers. |
1: Introduction
|
Lines 57-58: “For example, a single study of 20 mother child dyads vaccinated with CoronaVac (produced by Sinovac) has been published which showed no adverse effects in the breastfed children [9]”. The study has a very small sample size and the adverse events analyzed are not listed in the publication, hence how is this statement accurate? |
Dear Reviewer, thank you for the comment. The aim of quoting this study was to strengthen our argument that there is a paucity of research in the reactogenicity of non-mRNA vaccines specifically, which is stated before and after the quotation of this study. |
1. Introduction |
How did the authors assess adverse events (AEs) in their cohort? Did they provide a list of AEs and were the participants free to report AEs not listed on the survey? |
Dear Reviewer, thank you for the comment. The adverse events listed in our questionnaire are as follows: no symptoms, reaction at injection site, fatigue or low mood, headache, elevated body temperature upto 38C, soreness and enlarged lymph nodes, nausea, fever over 38C, runny nose, diarrhea, chest pain, cough, vomiting, allergic symptoms and others. To avoid survey fatigue, we did not list any other symptoms and opted to allow for free text submission. 2 Independent reviewers sifted through these responses and further categorized these “other” symptoms which are explained in section 3.2.1.
For completeness, we have included an Appendix which has our detailed questionnaire. |
Section 3.2.1 |
Others have shown that COVID vaccines induce higher antibody titers that cross the placental barrier and are found in umbilical cord blood (https://www.medrxiv.org/content/10.1101/2021.03.07.21253094v1). The effect of these antibodies cannot be ascertained in a few days or weeks. |
Dear Reviewer, thank you for the comment. We agree that COVID vaccines induce higher antibody titers transplacentally. Vaccination of lactating mothers serves as another layer of protection for the vulnerable infant through sIgA transfer of milk antibodies. |
N/A |
In fact, a similar study conducted in the US PMID: 34492204 seems to have similar observations as the authors present here, hence it would be nice if the authors can highlight the differences and uniqueness of their study. |
Dear Reviewer, thank you for the comment. We have included a section highlighting the differences and uniqueness of this study (US PMID: 34492204) compared to ours as seen in the Discussion section.
|
4. Discussion |
Lines 24-27: the sentence needs to be re-written. How can the majority experience both increase and decrease in lactation? |
Dear Reviewer, thank you for the feedback. To clarify our statement, we have amended this. It now reads: “Among the respondents who received non-mRNA COVID-19 vaccinations, the majority reported no change in lactation. “ |
Abstract |
It would be helpful to know exclusion and inclusion criteria. |
Dear Reviewer, thank you for the comment. We have added the exclusion and inclusion criteria.
This can be found under the section of “Materials and Methods” 2.3 Sample;
“Exclusion criteria included women who did not receive any WHO-approved COVID-19 vaccination, were not breastfeeding at the time of survey, could not provide consent in English, mentally disabled, or mothers of premature infants born <37 weeks gestation.” |
Materials and Methods; Section 2.3 |
It is unclear as to what questions were asked of the mothers? Was the data collected within 6 months of birth? Appendix A is nowhere to be found. |
Dear Reviewer, thank you for the comment. We have added Appendix A for your reference and we hope that this clarifies your query. Data was not necessarily collected within 6 months of birth and we included n=1,784 mothers with children aged 0 months to 39 months as long as they were still breastfeeding their child at the time of vaccination. |
Appendix A |

Reviewer 2 Report
I’ve read with interest the paper by Jacob-Chow and colleagues. The topic is interesting and the study confirms the safety of COVID-19 vaccines in lactating mother and baby dyads.
The study fits journal’s scope and general audience. However, there are some points that need further revision.
Overall, the paper is well written, but some spelling errors need to be amended.
It is not clear if authors evaluated previous SARS-CoV-2 infection or previous COVID-19. If they are too often used as synonym, are not the same and have not overlapping clinical significance.
The statistical analysis should further explore before considering this paper for publication. Multivariate analyses and models adjusting results for covariates and predictors should be added.
Limitations section is poor in consideration of the constitutive weaknesses of the research. Authors might want to consider the limitations due to the sample strategy, the type of study and the use of self-administered questionnaire, as well as the possible notoriety bias in AEFI reporting (also confirmed in pharmaco-epidemiologic studies of COVID-19 vaccines).
Author Response
Dear Reviewer,
Journal: Vaccines (ISSN 2076-393X)
Subject: Reactogenicity of mRNA and non-mRNA based COVID-19 vaccines among lactating mother and baby dyads” (Manuscript ID: vaccines-1746676)
We thank the reviewers for their comments and have amended the manuscript to address all their concerns accordingly. For ease of reference to the changes made, we have tabulated them below and highlighted them in our manuscript. We hope that the revised version will be suitable for publication. We look forward to hearing from you again.
Reviewer’s comment |
Response |
Section |
Overall, the paper is well written, but some spelling errors need to be amended. |
Dear Reviewer, thank you for the comment. We have re-conducted a full spell check on our paper, following British English spelling. |
N/A |
It is not clear if authors evaluated previous SARS-CoV-2 infection or previous COVID-19. If they are too often used as synonym, are not the same and have not overlapping clinical significance. |
Dear Reviewer, thank you for the comment. We acknowledge that: ● SARS-CoV2 is one of our key terms in the abstract, ● Two of our references (2,9) mention “SARS-CoV2”.
We took great care to avoid interchanging these terms and our paper only utilizes COVID-19 as a term in the full text. |
N/A |
The statistical analysis should further explore before considering this paper for publication. Multivariate analyses and models adjusting results for covariates and predictors should be added. |
Dear Reviewer, thank you for the comment. We have included such analysis throughout and our results were statistically significant. |
N/A |
Limitations section is poor in consideration of the constitutive weaknesses of the research. Authors might want to consider the limitations due to the sample strategy, the type of study and the use of self-administered questionnaires, as well as the possible notoriety bias in AEFI reporting (also confirmed in pharmaco-epidemiologic studies of COVID-19 vaccines). |
Dear Reviewer, thank you for the helpful comments. We have reviewed the literature as you have recommended and included the stated limitations in section 4.
We have amended this to read: “Limitations in our study include subjective reports from study participants; for example, there was no measurement of milk volume to quantify any perceived change in milk supply. Participants’ reported symptoms and signs were not verified by healthcare professionals. As with self-reported surveys, there is a possibility of recall bias resulting in over- or underestimation of the events reported. Lastly, there is a possibility of notoriety bias - in fact, various cognitive biases have been established in studies that focus on AEFI reporting (adverse effects following immunisation).” |
Discussion, paragraph 6 |

Round 2
Reviewer 1 Report
The authors have attempted to revise the manuscript in response to the comments provided. Most comments were addressed but some critical ones remain.
Abstract:
The most common maternal adverse reaction was a local reaction at the injection site; the largest minority of respondents, 49.6% (780/1571) , reported experiencing worse symptoms when receiving the second dose compared to the first dose. There were no major reported adverse effects or behavioral changes in the breastfed infants [for the duration of the study period]. Among the respondents who received non-mRNA COVID-19 vaccinations, a majority reported no change in lactation but those who did more commonly reported either an increase in milk supply, decrease in milk supply and pain in the breast. Among the respondents who received non-mRNA COVID-19 vaccinations, the majority (147/227, 64.8%) [hanging sentence- needs to be completed]. The more commonly reported lactation changes (fluctuations in breastmilk supply and pain in the breast) for the non-mRNA vaccines were similar to that of respondents who received mRNA vaccines. Our study, with a large cohort and racial mix, further augments earlier reported findings that COVID-19 vaccines are safe for breastfeeding mothers and her children in short term.
I do not agree with the last sentence- perhaps the authors can consider something like- Our study of racially diverse cohort of breastfeeding mothers suggests that COVID-19 vaccines did not cause any serious adverse events for the duration of the study in which the participants were surveyed, but long-term safety remains to be determined.
I thank the authors for attaching the survey questionnaire. It is unclear what behavioral changes in infants were assessed? The questionnaire did not really assess behavior and hence that sections needs to be revised.
The authors responded “Due to the nature of our study design, no control group was added. This is because based on various international guidelines, including WHO, it is recommended that lactating mothers receive COVID-19 vaccinations for immunity. Hence making it ethically unfeasible for a control group to be used.
No, it is not unethical to have an unvaccinated control group. Please be careful in making such assumptions and statements. Majority of the vaccines are under emergency use authorizations and as of today, most countries have lifted mandates, or a large number of countries did not ever require vaccinations. Vaccine trials in pregnant women were confined to a very small number of women and we are still learning about these vaccines. Hence it is presumptive to make the conclusion that vaccines are safe without a proper control group. The authors must tone down this rhetoric. If the authors have data from breastfeeding unvaccinated mothers, it will be a very important group to include and compare.
The authors state “Although children are mostly asymptomatic or have mild COVID-19 infections, in some cases a more severe clinical picture has been described [1,2]. It is thus important to protect both mother-child dyads from COVID-19 infection as maternal vaccination during lactation leads to antibody transfer through breastmilk [3,4].”
He authors interchangeably use child and infants- where is the evidence that breastfed infants have suffered from severe COVID? And children who have had severe COVID had some comorbid conditions. Can the authors comment on why women who have had COVID, why are natural antibodies not good enough? For most infectious disease, the past wisdom was that mother’s natural immunity also could protect the infant, so why for COVID it cannot provide protection? What is the functional evidence that natural immunity is less important? Most evidence points that the natural immunity is more diverse and robust.
Between the two non-mRNA vaccines, Oxford-AstraZeneca was better in terms of adverse events experienced by mothers and lactational outcomes.
Do the authors mean that mothers vaccinated with AZ experiences less AEs and had better lactational outcomes?
Lastly, there is a possibility of notoriety bias- this terminology is unclear. Please explain.
Author Response
Dear Reviewer,
Journal: Vaccines (ISSN 2076-393X)
Subject: Reactogenicity of mRNA and non-mRNA based COVID-19 vaccines among lactating mother and baby dyads (Manuscript ID: vaccines-1746676)
We thank the reviewer for their comments and have amended the manuscript to address all their concerns accordingly. For ease of reference to the changes made, we have tabulated them below and highlighted them in our manuscript. We hope that the revised version will be suitable for publication. We look forward to hearing from you.
Reviewer’s comment |
Response |
Section |
Abstract:
The most common maternal adverse reaction was a local reaction at the injection site; the largest minority of respondents, 49.6% (780/1571), reported experiencing worse symptoms when receiving the second dose compared to the first dose. There were no major reported adverse effects or behavioral changes in the breastfed infants [for the duration of the study period]. Among the respondents who received non-mRNA COVID-19 vaccinations, a majority reported no change in lactation but those who did more commonly reported either an increase in milk supply, decrease in milk supply and pain in the breast. Among the respondents who received non-mRNA COVID-19 vaccinations, the majority (147/227, 64.8%) [hanging sentence- needs to be completed]. The more commonly reported lactation changes (fluctuations in breastmilk supply and pain in the breast) for the non-mRNA vaccines were similar to that of respondents who received mRNA vaccines. Our study, with a large cohort and racial mix, further augments earlier reported findings that COVID-19 vaccines are safe for breastfeeding mothers and her children in short term.
I do not agree with the last sentence- perhaps the authors can consider something like- Our study of racially diverse cohort of breastfeeding mothers suggests that COVID-19 vaccines did not cause any serious adverse events for the duration of the study in which the participants were surveyed, but long-term safety remains to be determined. |
Dear Reviewer, thank you for the comment. We have amended the last line such that it reflects the limitation of our study in terms of long-term findings and included the other suggestions for specificity and completeness.
The last line now reads: “Our study, with a large, racially diverse cohort, further augments earlier reported findings that the COVID-19 vaccines tested in this study did not cause any serious adverse events in our population for the duration of the study in which participants were surveyed, although long-term effects have yet to be studied.” |
Abstract |
I thank the authors for attaching the survey questionnaire. It is unclear what behavioral changes in infants were assessed? The questionnaire did not really assess behavior and hence that sections needs to be revised. |
Dear Reviewer, thank you for the comment. We have amended the Results section to include a more detailed explanation of the means by which we assessed behaviour through the questionnaire.
The explanation is as follows: “Mothers were also asked to describe any other possible symptoms, including changes in behaviour such as increased fussiness or tiredness, they observed in their breastfed children following their COVID-19 vaccination doses; these results were then manually coded and categorised to generate the data shown in Table 2.” |
3.2.3. Impact on Child |
The authors responded “Due to the nature of our study design, no control group was added. This is because based on various international guidelines, including WHO, it is recommended that lactating mothers receive COVID-19 vaccinations for immunity. Hence making it ethically unfeasible for a control group to be used.”
No, it is not unethical to have an unvaccinated control group. Please be careful in making such assumptions and statements. Majority of the vaccines are under emergency use authorizations and as of today, most countries have lifted mandates, or a large number of countries did not ever require vaccinations. Vaccine trials in pregnant women were confined to a very small number of women and we are still learning about these vaccines. Hence it is presumptive to make the conclusion that vaccines are safe without a proper control group. The authors must tone down this rhetoric. If the authors have data from breastfeeding unvaccinated mothers, it will be a very important group to include and compare. |
Dear Reviewer, thank you for the constructive feedback. We acknowledge that this is a limitation of our study and future studies should include a control group for better comparison. |
Discussion |
The authors state “Although children are mostly asymptomatic or have mild COVID-19 infections, in some cases a more severe clinical picture has been described [1,2]. It is thus important to protect both mother-child dyads from COVID-19 infection as maternal vaccination during lactation leads to antibody transfer through breastmilk [3,4].”
He authors interchangeably use child and infants- where is the evidence that breastfed infants have suffered from severe COVID? And children who have had severe COVID had some comorbid conditions. Can the authors comment on why women who have had COVID, why are natural antibodies not good enough? For most infectious disease, the past wisdom was that mother’s natural immunity also could protect the infant, so why for COVID it cannot provide protection? What is the functional evidence that natural immunity is less important? Most evidence points that the natural immunity is more diverse and robust. |
Dear Reviewer, thank you for the comment and highlighting this as a concern.
We acknowledge this is a potentially contentious area as we continue to learn about COVID-19. We formed our argument based on several recent publications. For example, Pilz et al. suggested that hybrid immunity confers greatest protection against SARS-CoV-2 infection compared to natural immunity. Similarly, Jiang et al. suggested that newborns and infants are more susceptible to severe disease from a SARS-CoV-2 infection.
We would like to invite you to read the following paper (reference #6 – Pilz et al. SARS-CoV-2 reinfections: Overview of efficacy and duration of natural and hybrid immunity. Environ Res 2022.) which suggested that hybrid immunity appears to confer the greatest protection against SARS-CoV-2 infections, compared to natural immunity. We refer to their findings that “Data on the efficacy of hybrid immunity are inconsistent but point into the direction of hybrid immunity being superior as compared to either vaccine-induced (without a booster) or natural immunity alone.”
We would also like to invite you to read the following papers (references #1 – Jiang et al. COVID-19 and multisystem inflammatory syndrome in children and adolescents. The Lancet Infectious Diseases 2020, 20(11), pp. 276-e288.; and #2 – Bixler et al. SARS-CoV-2–Associated Deaths Among Persons Aged <21 Years — United States, February 12–July 31, 2020. MMWR. Morbidity and Mortality Weekly Report 2020, 69(37), pp.1324-1329.) for evidence regarding the susceptibility of newborns and infants to SARS-CoV-2 infections. We refer to their findings that “There is a U-shaped curve of severity in children diagnosed with COVID-19, and babies younger than 1 year are at a higher risk of developing severe COVID-19, although these infections are infrequent.”
We have also amended our paper for a more consistent use of the term “child” as opposed to “infant”.
|
Introduction |
“Between the two non-mRNA vaccines, Oxford-AstraZeneca was better in terms of adverse events experienced by mothers and lactational outcomes.”
Do the authors mean that mothers vaccinated with AZ experiences less AEs and had better lactational outcomes? |
Dear Reviewer, thank you for the comment.
We have rephrased the sentence for clarity. It has been amended to: “Between the two non-mRNA vaccines studied in this paper, Oxford-AstraZeneca was associated with fewer adverse events and overall better lactational outcomes.” |
Discussion |
“Lastly, there is a possibility of notoriety bias”
This terminology is unclear. Please explain. |
Dear Reviewer, thank you for the comment. We have amended the manuscript to include an explanation of the term and its relevance to our paper.
The sentence has been changed to: “There is also a possibility of notoriety bias, a type of selection bias, affecting adverse event reporting; our respondents’ awareness of media coverage and public opinions of adverse events following COVID-19 vaccinations – which were prominent and widely-discussed during the duration of this study – may have resulted in increased rates of reports of adverse events among our population.” |
Discussion |

Reviewer 2 Report
An issue still remains. Authors that only utilize COVID-19 as a term to avoid interchanging these terms. This is wrong, also if some previous articles did that. As I previously said, SARS-CoV-2 infection and COVID-19 are not the same and have not overlapping clinical significance.
Author Response
Dear Reviewer,
Journal: Vaccines (ISSN 2076-393X)
Subject: Reactogenicity of mRNA and non-mRNA based COVID-19 vaccines among lactating mother and baby dyads (Manuscript ID: vaccines-1746676)
We thank the reviewer for their comments and have amended the manuscript to address all their concerns accordingly. For ease of reference to the changes made, we have tabulated them below and highlighted them in our manuscript. We hope that the revised version will be suitable for publication. We look forward to hearing from you.
Reviewer’s comment |
Response |
Section |
An issue still remains. Authors that only utilize COVID-19 as a term to avoid interchanging these terms. This is wrong, also if some previous articles did that. As I previously said, SARS-CoV-2 infection and COVID-19 are not the same and have not overlapping clinical significance. |
Dear Reviewer, thank you for the comment. We have amended the terminology used in our manuscript to clearly delineate between SARS-CoV-2 infections and COVID-19. |
N/A |
